# The Impact of Tourism on Ecosystem Services Value: A Spatio-Temporal Analysis Based on BRT and GWR Modeling

**Jun Liu** [1,2]**, Mengting Yue** [3,]*****, Yiming Liu** [4,]*****, Ding Wen** [5] **and Yun Tong** [6]

1 School of Tourism, Hubei University, Wuhan 430062, China; magicliu@hubu.edu.cn
2 Tourism Development Institute, Hubei University, Wuhan 430062, China
3 School of Urban and Regional Science, East China Normal University, Shanghai 200241, China
4 Key Laboratory of Tectonics and Petroleum Resources Ministry of Education, China University of Geoscience, Wuhan 430074, China
5 South China Institute of Environmental Sciences, Ministry of Ecology and Environment, Guangzhou 510655, China; wending@scies.org
6 School of Tourism, Hainan University, Haikou 570228, China; tongyun@hainanu.edu.cn
* Correspondence: 51193902035@stu.ecnu.edu.cn (M.Y.); liuyiming12@cug.edu.cn (Y.L.); Tel.: +86-158-7236-9015 (M.Y.)

**Abstract:** The healthy development of the ecosystem and tourism in destinations plays an essential role in sustainable development. Taking Shennongjia as an example, we analyzed the spatial–temporal variation in the ecosystem services value (ESV) and investigated the impacts of tourism on ESV and their spatial heterogeneity using the geographically weighted regression (GWR) and boosting regression tree (BRT) models. The results showed that (1) the types of ecosystem services (ESs) were dominated by climate regulation and biodiversity. The ESV increased from 3.358 billion yuan to 8.910 billion yuan from 2005 to 2018 and showed significant spatial divergence, maintaining a long-term distribution pattern of high in the center and low at the border. (2) The GWR and BRT results showed that the Distance to Scenic Spots (DSS) and the Distance to Residential Areas (DRA) are important factors influencing ESV, with the Distance to Hotels (DH) and the Distance to Roads (DR) having a relatively weak influence on ESV. (3) The influencing factors presented positive and negative effects, and the degree of influence has spatial heterogeneity. The DRA and DH inhibited the increase in ESV in nearby areas, while DR was the driving factor for increasing ESV. The assessment results of DSS vary according to the models.

**Keywords:** ecosystem services value; influencing factors of tourism; spatial heterogeneity; geographically weighted regression; boosting regression tree





## 1. Introduction

The 2030 Agenda for Sustainable Development sets forth the goals of No Poverty and Zero Hunger and emphasizes the protection, restoration, and promotion of sustainable use of terrestrial ecosystems. As one of the fastest-growing sectors, tourism generates nearly 10% of global employment and is a significant driver of regional poverty reduction [1,2]. However, tourism development also brings with it issues such as resource depletion, species loss, and environmental pollution, which have a significant negative impact on the sustainability of the natural environment [3,4]. Nearly 80% of the ESs used to support human activities have declined over the past 50 years, according to a report published by the Intergovernmental science-policy Platform on Biodiversity and Ecosystem Services [5]. Identifying the tourism factors affecting ESV and grasping the influencing trend is an urgent problem to be solved in the current environmental governance of the tourism industry.

As the first region in China to be awarded UNESCO Man and Biosphere Nature Reserve (1990), World Geopark (2013), World Natural Heritage (2016), and National Park (2016), Shennongjia has become a typical ecotourism destination due to its high ecological

value. Its industrial structure has continuously shifted from the primary industry to the tertiary industry. In 2018, Shennongjia withdrew from the national level as a poverty-stricken county, but its ecological environment has undergone obvious changes because of the rapid development of tourism. At the same time, since poor areas and natural resource wealth areas often have a high degree of overlap, the development of economies with tourism as the leading industry will be accompanied by more significant environmental threats, such as human interference from residents and tourists, as well as air pollution and soil erosion [6–13]. Therefore, taking Shennongjia as an example, exploring the impact of tourism on ESV will have strong impacts and implications on green sustainability in global tourism destinations.

However, the existing studies on tourism environmental impact mainly focus on the specific environmental problems brought by tourism and the assessment and application of tourism environmental carrying capacity and pay less attention to ESV [4,14–24]. Chen et al. explored the need for ESV in weighing land use allocation in the context of nature reserves [24]. Furthermore, they considered the application of ESV to study the environmental impacts of tourism in the Wulingyuan Scenic Area [25]. Li et al. judged the impact of tourism on ESV by analyzing land-use changes caused by tourism [26]. Church et al. attempted to incorporate sustainable tourism into ecosystem assessments by analyzing the tourism characteristics of sub-global assessments of ESs [27]. Although the above studies make an effort to link the tourism with ESV, they do not quantitatively explore the response of ESV to the tourism industry.

As a critical part of the Millennium Ecosystem Assessment, the ESV has gained broader attention from researchers [28–31]. Previous studies have used equivalence factors, functional values, and model assessments to evaluate various ESVs in regions, watersheds, and ecosystems, and have been used to guide environmental protection and management [30,32–40]. As ESV research continues to intensify, the exploration of ESV driving mechanisms is gradually increasing, and scenario simulation, econometric modeling, and machine learning methods have been introduced to measure the effects of natural and social factors on ESV [41–45]. Most studies have analyzed ESV from a global perspective. Although many innovative research methods have been introduced, they have failed to reveal the spatial heterogeneity of influencing factors on ESV from a local perspective. Additionally, the data selection is mainly based on the statistical yearbook, and there is a lack of research on the tourism sector.

Therefore, this study takes Shennongjia as an example and introduces spatial distance data using ArcGIS10.2 to investigate the role and spatial heterogeneity of the influence of tourism on ESV. Considering the possible spatial autocorrelation between variables, the assumption that the OLS residual terms are independent cannot be applied [46]. This study uses GWR to analyze the relationship between ESV and influencing factors. As a spatial analysis tool, GWR allows non-stationary data and can effectively solve the spatial autocorrelation problem of explanatory variables [47]. BRT is a self-learning method based on the classification regression tree algorithm, which combines the advantages of regression trees and boosting, eliminating the interactions between influencing factors [48]. It is used chiefly in ecological field research [49]. In this study, we aim to analyze and predict the impact of tourism on ESV through the comparative study of GWR and BRT and provide implications for the sustainable development of ecotourism destinations.

## 2. Theoretical Framework

Tourism-Based Social–Ecological System theory believes that man and nature are complex systems closely connected, disturbed, and driven by internal and external factors [50,51]. The impact of tourism on destinations mainly comes from tourists, residents, and the destination landscape [52,53]. The model (Figure 1) illustrates the impact mechanism of tourism on ESV.

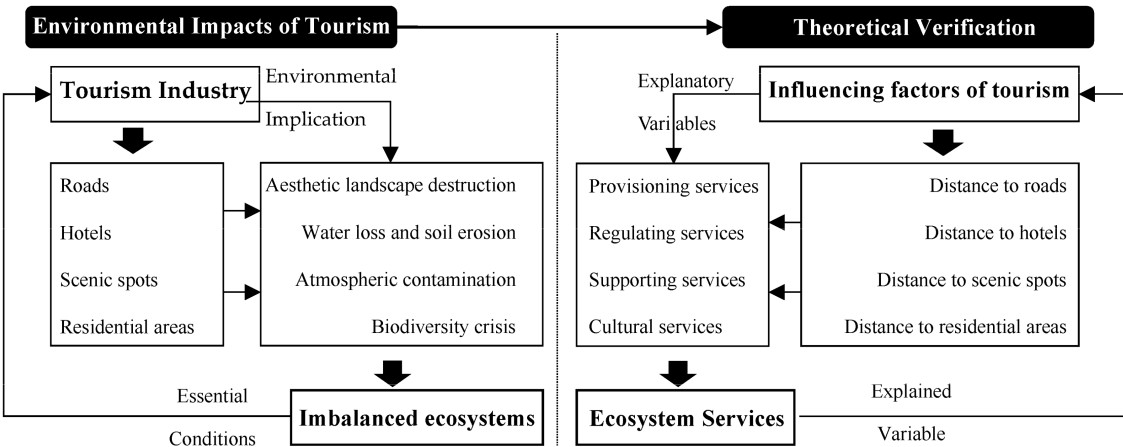

**Figure 1.** Mechanisms of tourism's impact on ESV.

As the primary source of tourism carbon emission, transportation is the main factor affecting the environment. Lin believes that tourists' different modes of transportation will directly affect carbon emissions [54]. Trombulak and Frissell argue that transport activities will harm biodiversity [55]. Nevertheless, there is also evidence supporting such a conclusion that proper transport infrastructure is good for the environment [56]. Community participation is key to developing ecotourism, and residents play an important role in tourism accommodation and hospitality. However, they may also have negative environmental impacts due to behaviors such as energy consumption and emissions [57]. The hotel industry's energy consumption and pollution have long been the focus of academic circles [58,59]. Some studies believe that the pollution of hotels will have an irreversible impact on the atmosphere and soil environment [60]. The construction of scenic spots is a critical project of tourism development. Its activities will change the land use type and landscape pattern of the tourist destination. The activities of tourists in the scenic spot will also interfere with the ecological environment [25].

In this paper, we attempt to establish the impact mechanism of tourism on ESV. We analyze the impact of hotels, transportation, residents, and scenic spots on ESV based on the actual situation of Shennongjia and the availability of data.

## 3. Materials and Methods

### 3.1. Study Area

Located in the western part of Hubei Province, China (Figure 2), Shennongjia (31°21'–31°36' N, 110°03'–110°33' E) covers an area of 3232.77 km², with six towns and two townships under its jurisdiction. The forest coverage in the region is over 90%, and it is a specific region with ecotourism as the leading industry. From 2005 to 2018, the number of tourism receptions and tourism revenue increased from 860,000 person-times and 180 million yuan to 15.86 million person-times and 5.73 billion yuan, respectively. The GDP increased nearly six fold, realizing the transformation from the uneven development mainly of farming and animal husbandry to tourism-led development. The ESV has also changed drastically under the multiple influences of ecological constraints, economic growth, and tourism industry development.

### 3.2. Methods

#### 3.2.1. ESV Assessment

We adopted the method proposed by Costanza et al. and Xie et al. to measure the ESV of Shennongjia [32,61]. However, due to the differences in the classification of land-use types, we reclassified the criteria of land-use types according to the literature [62]. Some of the ESs' equivalent values per unit area were directly adopted from Xie et al., including cropland, forests, water, and unutilized land [62]. Others were adjusted appropriately.

The equivalent value per unit area of high-coverage grasslands was calculated by the arithmetic mean taken from grassland, meadow, and shrub [63]. In accordance with the standard of high-coverage grassland, the ESs' equivalent values per unit area of moderate- and low-coverage grasslands were reduced in proper proportions [64]. Urban land, rural residential land, and other construction land were all classified as construction land and thus were not included in this assessment [65]. In this study, CPI was used to measure ESV accurately [66]. The methods are shown in Equations (1)–(5), where $E$ represents the ESs' equivalent value per unit area; $l$ is the area of the main crops in each year (ha); $q$ is the output of the main crops in each year (kg); $r$ is the number of main crop types; $I$ is the type of main crop, including indica rice, japonica rice, wheat, and maize; and $p_i$ is the unit price of the $i$-th crop type in each year (yuan/kg); $n$ is the number of land -use types; $A_m$ is the area of the m-th land -use type (ha); and $VC_m$ is the ESV in the unit area of the $m$-th land -use type (yuan/ha).

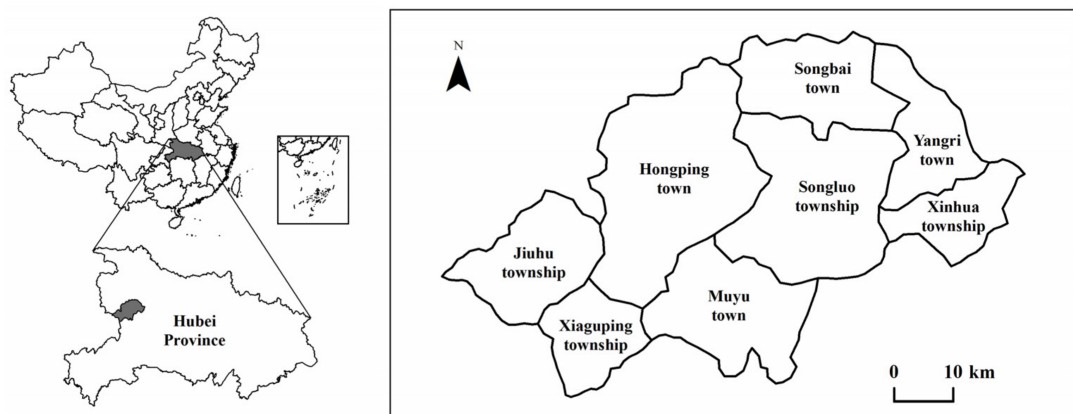

**Figure 2.** Location of the study area.

$$a_j = \frac{CPI_j}{100} \, (j = 2005, 2010, 2015, 2018) \tag{1}$$

$$E = \frac{1}{7} \times \frac{l}{q} \times \frac{1}{r} \sum_{i=1}^{r} p_i \tag{2}$$

$$VC'_m = unitESV \times E \tag{3}$$

$$VC_m = VC'_m \times \frac{a_j}{a_{2005}} \tag{4}$$

$$ESV = \sum_{m=1}^{n} A_m VC_m \tag{5}$$

### 3.2.2. Exploratory Spatial Data Analysis

We used the Field Calculator in ArcGIS 10.2 to process the ESV calculated by Equations (1) to (5) into a 1 km×1 km grid data set, and its accounting equation is shown in Equation (6). $ESV_t$ is the ESV in grid-scale ($10^4$ yuan); $n$ refers to the number of land -use types in each grid; and $A_t$ is the grid area (km$^2$). Global Moran's I is used for spatial autocorrelation analysis in Equation (7), where $N$ is the number of grids; $W_{uv}$ is the spatial weight matrix between grid $u$ and grid $v$; $X_u$ is the $u$-th grids' ESV; and $\overline{X} = \frac{1}{N} \sum_{u=1}^{n} X_u$.

$$ESV_t = \frac{1}{n} \sum_{m}^{n} VC_m \times A_t \tag{6}$$

$$I = \frac{N}{\sum_{u=1}^{N} \sum_{v=1}^{N} W_{uv}} \frac{\sum_{u=1}^{N} \sum_{v=1}^{N} W_{uv} (X_u - \overline{X})(X_v - \overline{X})}{\sum_{u=1}^{N} (X_u - \overline{X})^2} \tag{7}$$

### 3.2.3. Geographically Weighted Regression (GWR)

By estimating local parameters as a proxy for global parameters, GWR is used to explore the strength of influence and spatial heterogeneity of explanatory variables at multiple scales [67]. The equation is as follows, where $y_i$ is the value at sampling point $i$ (namely unit grid ESV); $\beta_0$ represents the intercept; $(\mu_i, v_i)$ is the projection coordinate of sampling point i; $\beta_k(\mu_i, v_i)$ is the coefficient of the $k$-th independent variable at sampling point i; $x_{ik}$ is the $k$-th independent variable at sampling point i; and $\varepsilon_i$ is the random error term.

$$y_i = \beta_0(u_i, v_i) + \sum_{k=1}^{p} \beta_k(u_i, v_i)x_{ik} + \varepsilon_i \tag{8}$$

### 3.2.4. Boosting Regression Tree (BRT)

With a randomly selected data set, BRT analyzes the degree of influence of the independent variable on the dependent variable, and the remaining data in the sample are used to validate the accuracy of the model in the form of recursive binary partitioning to achieve an iterative fit, which is mainly used to predict the contribution of influencing factors and their spatial heterogeneity [68]. We used the gbm package developed by Elith et al. (2008) in R for modeling, set the learning rate to 0.005 regarding similar studies, extracted 50% of the data for regression analysis, and performed five cross-validations [69,70].

### 3.2.5. Selection of Influencing Factors

The existing research on tourism environmental impact mainly adopts descriptive analysis, questionnaire survey and interview, statistical analysis, mathematical modeling, and on-site sampling to carry out quantitative or qualitative research [19,21,22,71–77]. Here, we discuss the spatial heterogeneity of influencing factors from the perspective of distance using spatial data. Considering the characteristics of ecotourism in poor areas, we hypothesized that the impacts of tourism on the environment are mainly reflected in tourism development and tourists' and residents' activities, and we used seven independent variables in two dimensions to explore the impact of tourism on ESV, specifically including:

(1) Distance to Hotels (DH): An indicator to investigate the impact of hotel distribution on ESV. On the one hand, hotels harm the environment through energy consumption and emissions [58,76]; on the other hand, the level of hotel revenue will affect government revenue, which will affect environmental protection investment and ecological protection [77].

(2) Distance to Scenic Spots (DSS): An indicator to explore the impact of scenic spots' distribution on ESV. The construction of scenic spots limits the harmful disturbance of human activities to strictly protected areas but disrupts the balance of ecosystems in the visitor-accessible areas [78].

(3) Distance to Roads (DR): An indicator to explore the impact of road distribution on ESV. The construction of roads improves the accessibility of the area. Its environmental protection effect is achieved by replacing private transport with public transport, reducing road congestion, and reducing air pollution caused by traffic congestion [79]. However, the impact of tourism traffic on the soil environment and vegetation diversity around the roads cannot be ignored [19]. Additionally, the transportation investments will also indirectly affect the expenditure of environmental protection projects, which affects the effectiveness of environmental protection [80].

(4) Distance to Residential Areas (DRA): An indicator used to describe the distance from the residential areas. As the prominent place that hosts the activities of residents, the settlement becomes a tourist accommodation and catering reception area in addition to hotels. DRA is used as an indicator to explore the impact of residential areas on ESV.

(5) Referring to Li et al., DEM, slope, and aspect were selected as control variables [70].

### 3.2.6. Data Sources and Processing

The data sources in this paper are mainly divided into three categories. (1) Satellite image data sets: including land use data and digital elevation models (DEMs). The land use data are decoded from Landsat-TM/ETM and Landsat OLI remote sensing image data as the primary information sources, with an overall accuracy of 88.95%, which meets the research requirements. The data are from the Resource and Environment Science Data Center of the Chinese Academy of Sciences (http://www.resdc.cn/, accessed on 5 January 2021). (2) Statistical data set: data such as CPI and food production value used to evaluate ESV are based on *Shennongjia Forestry District Statistical Yearbook* and *China Agricultural Products Price Survey Yearbook*. (3) Spatial data set: including the geographic location information of hotels, residential areas, roads, and scenic spots. The data of hotels are provided by crawling the data of the eLong website (http://hotel.elong.com/, accessed on 5 January 2021), the data of residential areas are obtained by extracting the land use attribute data through ArcGIS, and the data of roads come from OpenStreetMap website (http://download.geofabrik.de/asia/, accessed on 5 January 2021). Scenic spots are collected from retrieving the longitude and latitude distribution of scenic area through Google Maps (https://www.google.com/maps/, accessed on 5 January 2021).

Due to the different data types required by GWR and BRT, we used ArcGIS to preprocess the data (Figure 3). Firstly, with the help of the 3D Analyst-Raster Surface tool, DEM was converted into slope and aspect data. The data of hotels, roads, residential areas, and scenic spots are derived using the Euclidean Distance tool. The ESV data are calculated by Equation (6). Secondly, we transformed the above data into a 1 km × 1 km raster layer and used the Extract Multi Values to Points tool to extract the property values to 3270 randomly generated vector Points. Thirdly, we used the Jenks natural breakpoint method to convert the data set from continuous to discrete variables to comply with the GWR requirements. Finally, we overlaid the grid ESV spatial distribution diagrams of 2005, 2010, 2015, and 2018 in pairs. If the ESV changed, the value was assigned to 1. If the ESV had not changed, the value was assigned to 0. The explained variable was binarized through the above steps to meet the requirements of BRT.

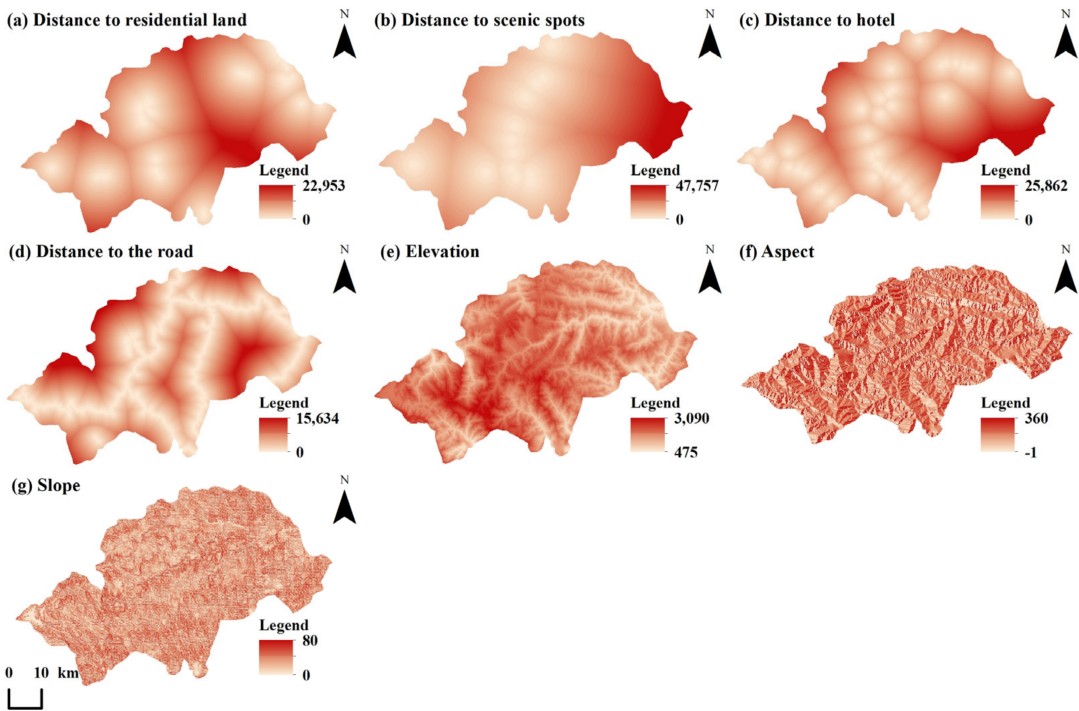

**Figure 3.** Spatial distribution map of the factors in 2018.

## 4. Results

### 4.1. Spatial–Temporal Variation of ESV

The estimation results of ESV in 2005, 2010, 2015, and 2018 are 3.358 billion yuan, 5.385 billion yuan, 7.774 billion yuan, and 8.910 billion yuan, respectively, with a total added value of 5.552 billion yuan, according to Table 1. The ESV of 11 sub-types all showed a steady growth trend. The functions of environment depuration, nutrient cycle maintenance, and biodiversity maintained a long-term growth at the same rate. In contrast, the growth rates of food production, raw material production, gas regulation, climate regulation, and soil conservation functions decreased. With the strengthening of environmental protection and tourism development, the growth rate of water supply, hydrological adjusting, and aesthetic landscape function has increased continuously. The ESV of the above functions has increased from 48 million yuan, 643 million yuan, and 158 million yuan to 128 million yuan, 1.712 billion yuan, and 418 million yuan, respectively, and the proportion of the ESV is generally stable at about 4–5%.

**Table 1.** Changes in 11 sub-types of ESV from 2005 to 2018 (unit: billion yuan).

| ESV | | 2005 | | 2010 | | 2015 | | 2018 | |
|---|---|---|---|---|---|---|---|---|---|
| | | ESV | Proportion/% | ESV | Proportion/% | ESV | Proportion/% | ESV | Proportion/% |
| Provisioning services | Food production | 0.48 | 1.44 | 0.77 | 1.43 | 1.11 | 1.43 | 1.27 | 1.43 |
| | Raw material production | 1.00 | 2.97 | 1.59 | 2.96 | 2.30 | 2.96 | 2.64 | 2.96 |
| | Water supply | 0.48 | 1.43 | 0.78 | 1.44 | 1.12 | 1.44 | 1.28 | 1.44 |
| Regulating services | Gas regulation | 3.27 | 9.74 | 5.24 | 9.73 | 7.56 | 9.73 | 8.67 | 9.73 |
| | Climate regulation | 9.63 | 28.67 | 15.43 | 28.65 | 22.27 | 28.65 | 25.53 | 28.65 |
| | Environment depuration | 2.84 | 8.45 | 4.55 | 8.45 | 6.57 | 8.45 | 7.53 | 8.45 |
| | Hydrological adjusting | 6.43 | 19.15 | 10.35 | 19.22 | 14.94 | 19.22 | 17.12 | 19.21 |
| | Soil conservation | 3.98 | 11.85 | 6.37 | 11.84 | 9.20 | 11.83 | 10.55 | 11.84 |
| Supporting services | Nutrients cycle maintenance | 0.30 | 0.90 | 0.48 | 0.90 | 0.70 | 0.90 | 0.80 | 0.90 |
| | Biodiversity | 3.59 | 10.69 | 5.76 | 10.69 | 8.31 | 10.69 | 9.52 | 10.69 |
| Cultural services | Aesthetic landscape | 1.58 | 4.69 | 2.53 | 4.70 | 3.65 | 4.70 | 4.18 | 4.70 |
| Total | | 33.58 | 100 | 53.85 | 100 | 77.74 | 100 | 89.10 | 100 |

Since the spatial distribution and change of ESV were steady, this paper only provided the spatial distribution of ESV in 2005 and 2008 and incremental changes in ESV from 2005 to 2010 and from 2010 to 2015 (Figure 4). The high-value area was mainly distributed in the Shennongjia National Nature Reserve, while the low-value area was mainly located in the eastern boundary. The spatial variation in ESV increments was consistent with ESV stock, which showed that the ESV increment in the nature reserve continued to increase considerably. In contrast, the ESV increment in the eastern and western Shennongjia was relatively tiny. In general, ESV shows a rapidly rising trend, and its temporal and spatial evolution state is relatively stable.

The results of Global Moran's I were all 0.000, which passed the significance test at the 1% level, indicating that the ESV had a significant high (low) agglomeration spatial autocorrelation. The results also showed that it is reasonable to use GWR to carry out the regression of influencing factors.

### 4.2. The Impact of Tourism Factors on ESV

#### 4.2.1. Fitting Results of GWR

Because of the large number of dependent variables and the tendency to bias the regression coefficients due to mutual influence, we used SPSS 23 to test the cointegration of the dependent variables. The results (Table 2) show that the VIF was less than ten. The conditional index was less than 30, indicating no covariance between the indicators.

The proportion of the positive and negative values of the regression coefficients and the results of the coefficients indicated that the explanatory variables have different effects on ESV (Table 3). DH and DRA mainly have adverse effects, while DSS and DR primarily positively affect ESV. From the perspective of spatiotemporal changes, the areas covered by the positive impact of DH and DR are increasing. In contrast, the areas covered by the

positive impact of DSS are continuously decreasing. The proportion of the areas positively affected by DRA showed a V-shaped change trend, and the coverage proportion of the positive impact expanded overall.

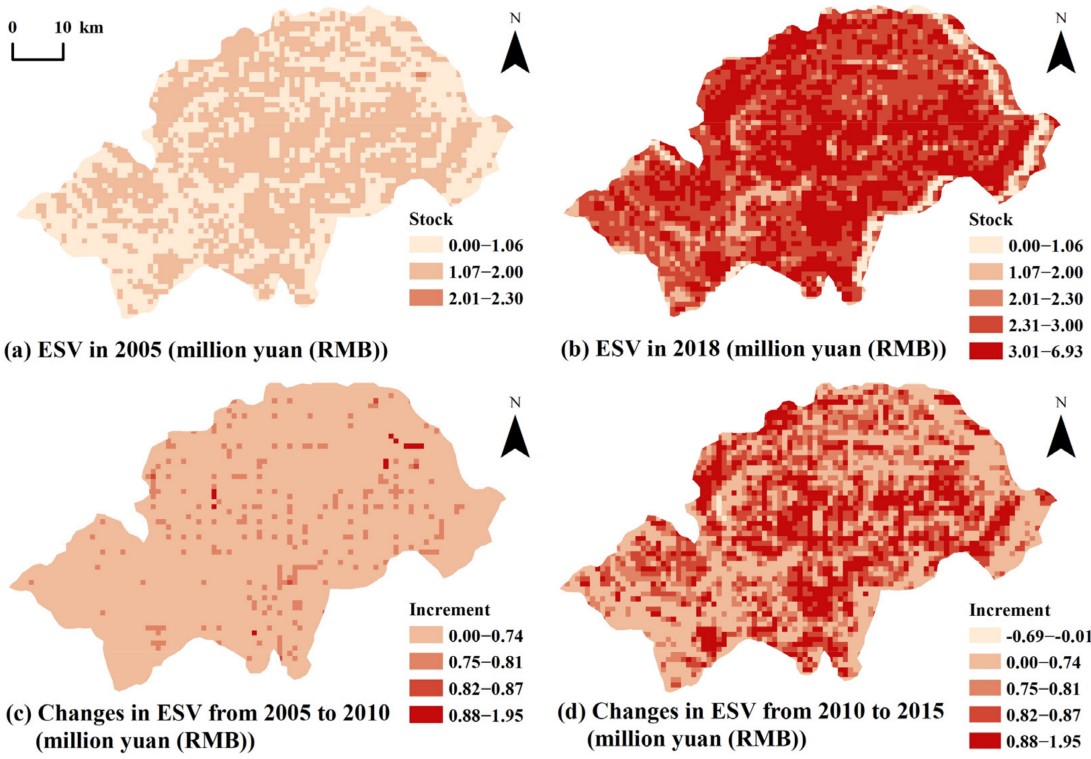

**Figure 4.** Spatial distribution of ESV and its increments.

**Table 2.** Collinearity inspection of the influencing factors.

| Factors | 2010 | | | 2015 | | | 2018 | | |
|---------|------|-----------|------|------|-----------|------|------|-----------|------|
| | VIF | Tolerance | CI | VIF | Tolerance | CI | VIF | Tolerance | CI |
| DH | 2.95 | 0.34 | 3.35 | 3.31 | 0.30 | 3.40 | 2.79 | 0.36 | 3.38 |
| DSS | 2.17 | 0.46 | 3.94 | 2.56 | 0.39 | 4.12 | 2.74 | 0.37 | 4.34 |
| DR | 1.04 | 0.96 | 5.16 | 1.08 | 0.93 | 5.13 | 1.27 | 0.79 | 5.04 |
| DRA | 2.09 | 0.48 | 5.82 | 1.97 | 0.51 | 5.73 | 1.23 | 0.81 | 5.37 |
| DEM | 1.35 | 0.74 | 7.39 | 1.29 | 0.78 | 7.61 | 1.39 | 0.72 | 6.97 |
| Slope | 1.03 | 0.97 | 10.63 | 1.02 | 0.98 | 10.07 | 1.01 | 0.99 | 8.34 |
| Aspect | 1.01 | 0.99 | 17.71 | 1.01 | 0.99 | 17.25 | 1.01 | 0.99 | 17.77 |

Note: VIF, variance inflation factor; CI, condition indices.

**Table 3.** Calculation results of GWR.

| Explanatory Variables | 2010 | | | 2015 | | | 2018 | | |
|-----------------------|---------|--------------------|--------------------|---------|--------------------|--------------------|---------|--------------------|--------------------|
| | Average | Positive Numbers | Negative Numbers | Average | Positive Numbers | Negative Numbers | Average | Positive Numbers | Negative Numbers |
| DH | 5.13 | 23.25% | 76.75% | 7.05 | 26.44% | 73.56% | 6.06 | 47.35% | 52.65% |
| DSS | 8.10 | 92.60% | 7.40% | 10.43 | 89.27% | 10.73% | 8.81 | 80.32% | 19.68% |
| DR | 3.98 | 81.16% | 18.84% | 7.43 | 77.45% | 22.55% | 6.85 | 85.58% | 14.42% |
| DRA | 6.00 | 16.09% | 83.91% | 8.01 | 9.89% | 90.11% | 12.07 | 23.08% | 76.92% |
| DEM | 7.83 | 98.27% | 1.73% | 12.05 | 98.63% | 1.37% | 10.51 | 94.50% | 5.50% |
| Slope | 2.63 | 96.23% | 3.77% | 3.44 | 92.63% | 7.37% | 3.82 | 92.66% | 7.34% |
| Aspect | 1.61 | 92.13% | 7.87% | 2.45 | 84.21% | 15.79% | 3.04 | 59.40% | 40.60% |
| Best Bandwidth | 28,299 | | | 19,829 | | | 14,580 | | |
| $R^2$ | 0.62 | | | 0.63 | | | 0.43 | | |
| Adjusted $R^2$ | 0.50 | | | 0.51 | | | 0.36 | | |

Note: The mean value in the table is the mean value after taking the absolute value of each regression coefficient.

### 4.2.2. Fitting Results of BRT

The ROC values of the training data were 0.978, 0.943, and 0.967, and the ROC values of the validation data were 0.874, 0.821, and 0.775, respectively. The ROC values of the three phases were above 0.75, indicating a high degree of model fit. By comparing the contribution rates of influencing factors in three stages (Table 4), it was found that DRA, DSS, and DH have the strongest influence on ESV, accounting for 48.5–60.9% in total. It was followed by slope and aspect (16.3–32.1%), DR (4.3–16.2%), and DEM (7.6–10.7%). In general, ESV is mainly affected by tourism; natural factors have a weak role in influencing ESV.

**Table 4.** Contribution rate of influencing factors.

| Explanatory Variables | 2005–2010 | | 2010–2015 | | 2015–2018 | |
| --- | --- | --- | --- | --- | --- | --- |
| | Contribution | Rankings | Contribution | Rankings | Contribution | Rankings |
| DRA | 20.1% | 1 | 23.3% | 1 | 19.6% | 1 |
| DSS | 19.9% | 2 | 23.1% | 2 | 15.1% | 3 |
| DH | 19.9% | 3 | 14.5% | 3 | 13.8% | 5 |
| DR | 16.2% | 4 | 4.3% | 7 | 10.5% | 6 |
| Slope | 10.2% | 5 | 13.4% | 4 | 14.8% | 4 |
| DEM | 7.6% | 6 | 10.7% | 6 | 8.8% | 7 |
| Aspect | 6.1% | 7 | 10.8% | 5 | 17.3% | 2 |

### 4.3. Spatial Heterogeneity of Tourism-Influencing Factors

This study focuses on the impact of tourism on ESV. Due to the limitation of space, we only analyzed the regression results of tourism-influencing factors.

#### 4.3.1. Impact of DRA on ESV

The influence of DRA on ESV shows noticeable spatial differences (Figure 5). The area near the residential areas has a negative effect, while the area far away has a positive effect. The regression coefficient of the eastern region is generally higher than that of the western region, and the intensity of the negative effect decreases continuously and weakens rapidly with the increasing distance. In contrast, the intensity of the positive effect is relatively stable.

#### 4.3.2. Impact of DSS on ESV

The spatial–temporal variation of the influence was stable and regular (Figure 6). The near-scenic area has a positive effect, while the far-sight area has a negative effect. The negative influence area gradually spreads from the middle to the south. Specifically, taking 11 km as the boundary, there is a negative influence within its range, and otherwise, there is a positive influence.

#### 4.3.3. Impact of DH on ESV

The GWR results showed that hotels had a negative impact on 67.65% of ESV (Figure 7). From 2010 to 2015, ESV in areas near hotels was negatively affected, while ESV in areas far away was positively affected. In 2018, however, the pattern was different. BRT results are contrary to GWR. The radiation range of the hotel's positive effect on ESV decreases from 13 km to 5 km, and its effect gradually turns from positive to negative.

#### 4.3.4. Impact of DR on ESV

The influence of DR on ESV fluctuates wildly (Figure 8), and the overall effect is positive with the increasing intensity. The radiation range of positive influence decreased from 20 km to 10 km, showing a circular contraction. It shows that the road construction in Shennongjia is beneficial to the healthy development of the ecosystem.

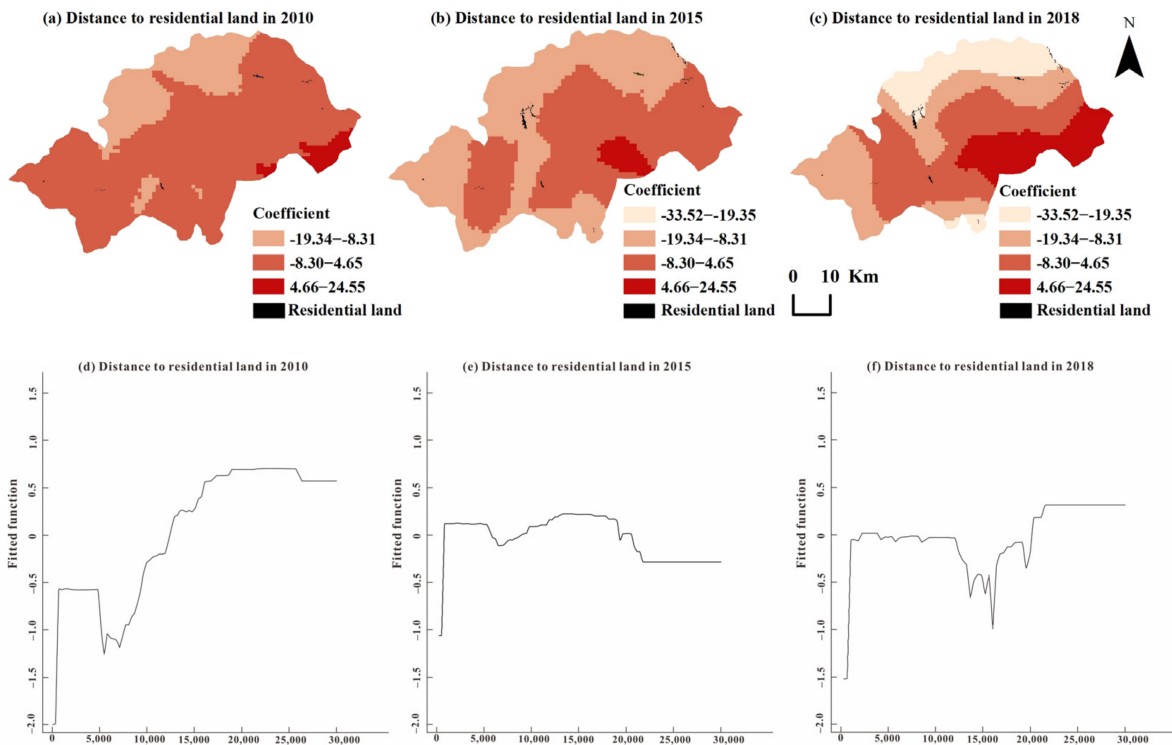

**Figure 5.** Influence of DRA on ESV based on GWR (**a**–**c**) and BRT (**d**–**f**).

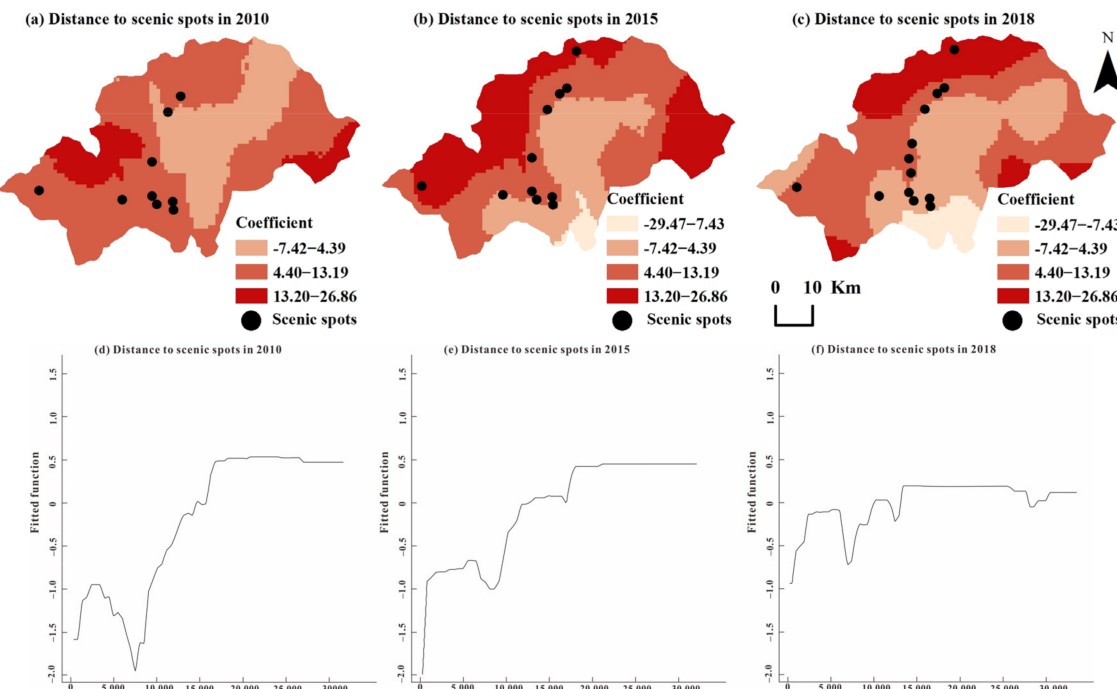

**Figure 6.** Influence of DSS on ESV based on GWR (**a**–**c**) and BRT (**d**–**f**).

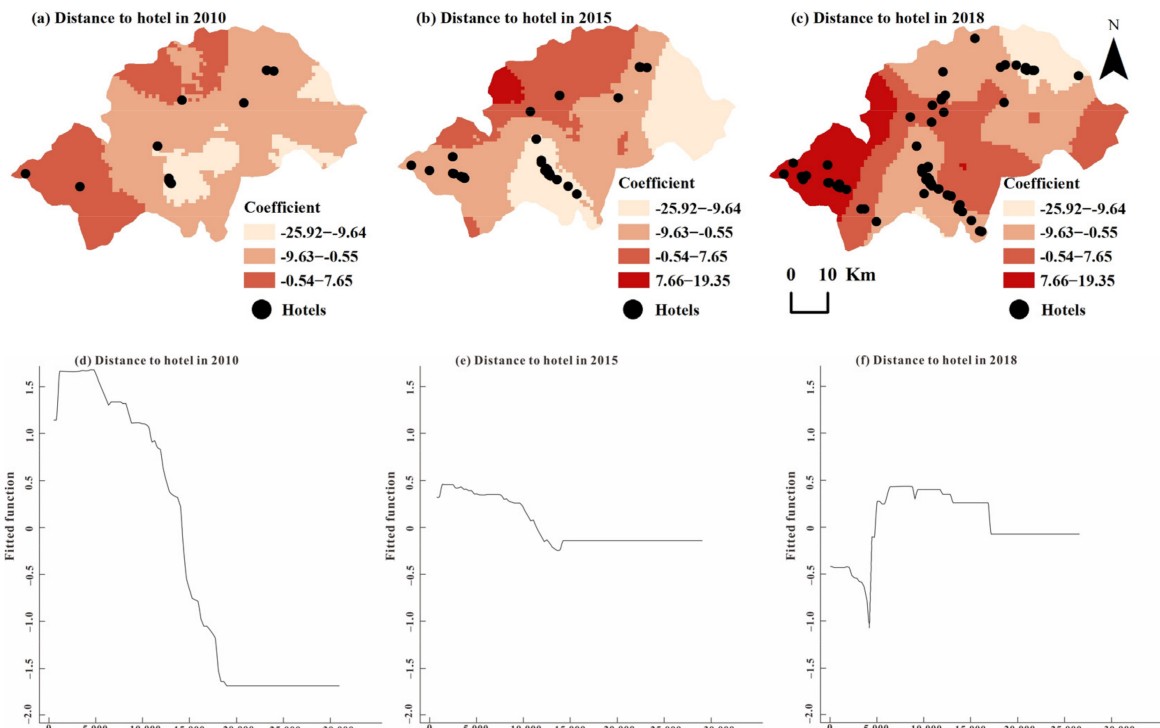

**Figure 7.** Influence of DH on ESV based on GWR (**a**–**c**) and BRT (**d**–**f**).

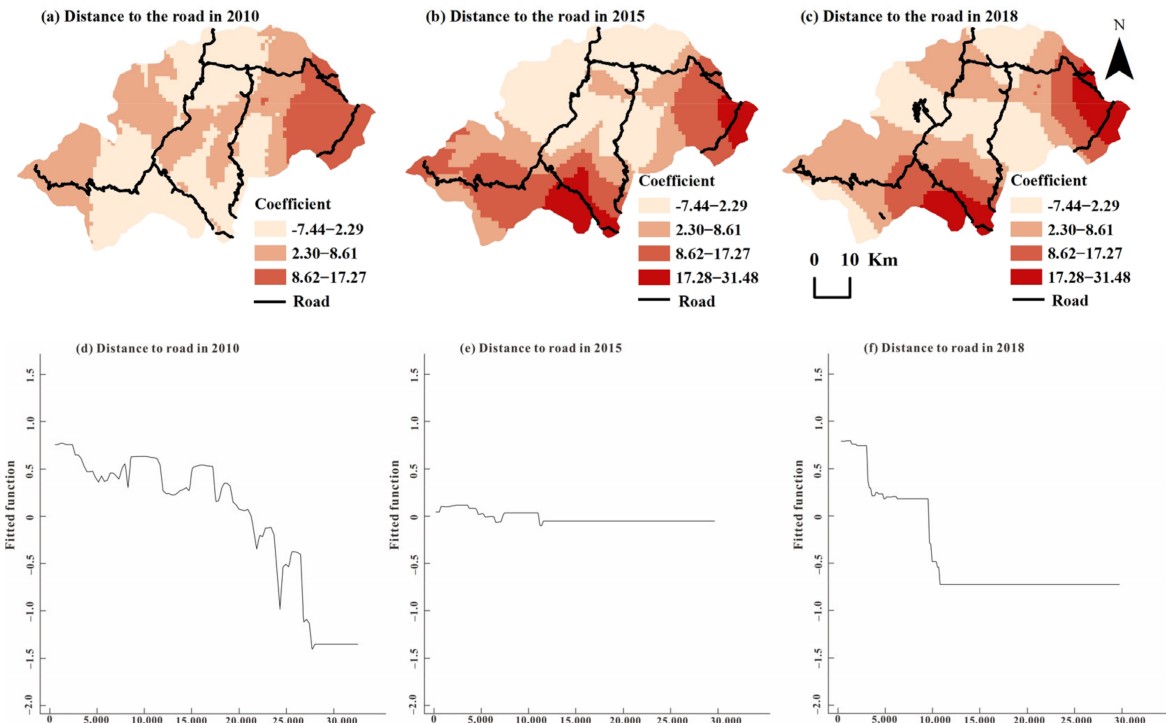

**Figure 8.** Influence of DR on ESV based on GWR (**a**–**c**) and BRT (**d**–**f**).

## 5. Discussion

### 5.1. ESV and Land Use Changes

The sustainable development of ecotourism destinations has been widely emphasized. This is mainly because the vulnerable destinations in China often lack the sustainable economic development model, but they have favorable tourism resources [81,82]. The rapid expansion of construction land and the acceleration of urbanization are recognized

as the reasons for ecological problems and the decrease in ESV [83]. In this paper, we used land use data to measure ESV and do not analyze land use change separately in the main text due to the limitation of research topic and space. In contrast to the findings of existing studies, we found that the ESV increased rather than decreased as the area of forest and cropland decreased and the area of construction land increased. This is mainly because the ESV per unit area of forest and cropland has increased. Compared with the land area, the ESV per unit area can significantly impact the ecosystem of forest and cropland. According to the ESV measurement process, based on specific land use types, the impact of ESV per unit area (*E* in Equation (3)) on ESV is far more significant than that of the area on ESV, which also confirms the necessity of using CPI to revise ESV [84]. This conclusion also reflects that the utilization efficiency of unused land is relatively high in tourism development and urbanization. Nevertheless, at the same time, it should be recognized that the development of tourism will inevitably bring about changes in land use types, and the government should prevent the large-scale occupation of ecological land in the process of tourism development.

### 5.2. Contribution of Influencing Factors Based on BRT and GWR

The GWR and BRT models selected for this study are complementary in assessing the factors influencing ESV. GWR supports parameter estimation of local changes in the association between the explanatory and explanatory variables and can spatially reflect neglected local characteristics; BRT is proficient in dealing with nonlinear interactions between explanatory variables and emphasizes its predictive function.

The regression results of GWR and BRT are consistent to a certain extent, both of which prove that the DSS and DRA are the dominant environmental impact factors. In contrast, the impact of DH and DR is weaker. However, due to the different model principles and algorithms, the environmental impact intensity of tourism measured by the two models shows the difference between time and space. The results of GWR show that the influence of tourism factors on ESV is ranked as follows: DSS, DRA, DR, and DH. According to the modeling results of BRT, its contribution rate is DRA, DSS, DH, and DR.

### 5.3. Spatial Heterogeneity of Influencing Factors Based on BRT and GWR

By comparing the role of tourism-influencing factors on ESV in the two types of models, we found that the direction and magnitude of the influence of DRA, DH, and DR behaved more consistently in time and space. The validation results for DSS contradicted each other.

DRA and ESV: The validation results of the two models both indicate that the construction of residential areas will have a negative impact on the environment in the immediate area and a positive impact on the remote area. As the direct stakeholders of tourism activities, residential areas collect a large amount of domestic wastewater, excreta, smoke, and other pollutants because they undertake the most tourism accommodation and catering activities [85]. This is consistent with the conclusion that while community residents eliminate poverty through natural resources and the tourism industry, they also reduce biodiversity due to excessive demand and abuse of nature [86].

DH and ESV: The results of GWR validate the negative impact effect of hotel proximity on ESV; the results of BRT conclude that hotel proximity mainly exhibits a positive impact effect from 2010 to 2015, but its intensity keeps decreasing and shows a negative impact effect in 2018. Overall, the validation results of the two models tend to be consistent. As the primary source of carbon emissions from tourism, the air pollution, soil disturbance, and vegetation damage caused by hotels have become increasingly prominent and intensified along with the mature development of tourism. The economic dividends brought by hotels cannot balance their negative impacts, leading to the negative impact on ESV in the neighboring areas.

DR and ESV: The validation results of both models show that DR has a positive impact on the near-area environment and negatively impacts the distant area. The conclusion of

this study supports the improvement effect. For Shennongjia, which is located in a poor mountainous area, the traffic block is the main hindrance of its rapid economic development. The increase in the length of the transportation route improves accessibility, which will contribute to the increase in tourist attraction and tourism revenue, thus providing sufficient funds for environmental protection.

DSS and ESV: The GWR modeling results show a positive environmental impact for near-scenic areas. In contrast, the BRT results suggest that DSS will have a negative impact on ESV in the surrounding areas. The results of GWR and BRT are also more contradictory for areas far from scenic areas. Nature reserves, national parks, and world heritage sites have always been critical natural resources for tourism development. The closure of strictly protected areas can effectively prevent human activities from disturbing flora, fauna, and soil. However, within a specific distance (such as places for tourists to visit and living areas for residents), it is impossible to avoid the environmental impact caused by tourism.

As noted in the literature review, exploring the environmental impact of tourism is critical to sustainable development. In this paper, we introduced GWR and BRT to verify the influence of tourism factors on ESV. We discussed the spatial heterogeneity of tourism-influencing factors, considering the natural and cultural values of the ecosystem. Previous studies only took the "point data" such as high-speed railway stations, bus stations, and airports as indicators in terms of research methods. In this research, the above traffic indicators were included with the help of "line data", reflecting the impact of private transportation on the environment, and improved the accuracy of conclusions. Although this paper has some innovations in research perspectives, selection of factors, and methods, and, through tourism environmental impact assessment, it reflects the degree of coordinated development of tourism economy and ecological protection to some extent, there is still room for expansion. First of all, the study failed to collect the data of tourists. After comparing with the statistical yearbook data, we found that the validity of the data obtained through Python crawlers was extremely low, and there was a lack of accurate tourism flow data in the required period. Secondly, due to the small research scale and low data availability, this paper is subject to certain constraints regarding universality and reference. Therefore, we will consider expanding the research scope in the future and carry out related studies in combination with ecological efficiency.

## 6. Conclusions

In this study, we explored the effects of tourism-influencing factors on ESV and underlined implications for the green and sustainable development of ecotourism in similar destinations. The results show that the ESV of Shennongjia has increased drastically. The proportion of 11 ESs in ESV is stable, and climate regulation, hydrological adjusting, soil conservation, and biodiversity constitute the main ecological functions. The ESV presents a high autocorrelation, showing a distribution pattern of high in the middle and low in the boundary. GWR results indicate that the intensity of tourism influencing factors is in the order of DSS, DRA, DR, and DH, while the results of BRT are DRA, DSS, DH, and DR. In terms of spatial heterogeneity, the results of GWR concluded that hotels and residents mostly negatively affected ESV, while scenic spots and roads mainly positively affected them. The results of BRT's spatial analysis of hotels, residents, and roads were consistent with GWR. For DSS, the results of BRT concluded that anthropogenic activities have a negative impact on the ecosystem in the near scenic area.

As pointed out in the discussion, although this paper has made some innovations in research objects and methods, there are still some limitations that must be further discussed. Firstly, as a typical influencing factor to be considered in tourism research, tourist data are not included in this study, which is mainly due to the low reliability and validity of the data. Secondly, this paper takes Shennongjia as an example. Although the tourism industry in Shennongjia is relatively mature and its research results have high reference value, it is limited by the scale of the scenic spots and provides little research value for other types of tourist destinations. In the future, we will consider expanding the research

scope and conducting more extensive empirical research combining different types of eco-tourism destinations in China to explore the universal law of tourism's impact on ecosystem services value.

**Author Contributions:** Formal analysis, Y.T.; Methodology, D.W.; Resources, Y.L.; Software, Y.L.; Writing—original draft, M.Y.; Writing—review and editing, J.L. All authors have read and agreed to the published version of the manuscript.

**Funding:** This work was supported by the National Social Science Foundation of China under grant no. 17CJY051.

**Institutional Review Board Statement:** Not applicable.

**Informed Consent Statement:** Not applicable.

**Data Availability Statement:** Not applicable.

**Conflicts of Interest:** The authors declare no conflict of interest.

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
