# Peer review of "The Impact of Tourism on Ecosystem Services Value: A Spatio-Temporal Analysis Based on BRT and GWR Modeling"

_sustainability, doi:10.3390/su14052587_

Round 1
Reviewer 1 Report
Review of “The impact of tourism on ecosystem services value: A spatial-temporal analysis based on BRT and GWR modeling”
Manuscript ID: sustainability-592479
The paper concerns the impacts of tourism on ecosystem services values (ESV). The assessment of factors that influence ESV was performed using two models: GWR – geographically weighted regression and BRT – boosting regression tree. The issue is analyzed in temporal and spatial dimensions on the example of Shennongjia which is a part of Hubei Province in China. The region has faced very intense economic growth, mainly due to tourism (ecotourism?) development and seems to be an interesting study area.
The layout of the paper and the research design is correct. I have just a few minor comments.
Comment 1: Abstract
When presenting the results concerning the increase of ESV you should refer to the specific year. In the abstract there is no information that you analyzed data for the years 2005-2018.
Comment 2: References
The article should be referenced in accordance with the editorial requirements. In the current version, for instance, you refer to Li et al. (2014) (lines 65, 68, 196…) and it is not clear which item it concerns (no 45 or 46 - lines 580-582).
Comment 3: Figures
I suggest to improve quality of some figures, especially Fig. 4.
Date 2nd. Of February 2022
Author Response
Response to Reviewer 1 Comments
Point 1: Abstract, When presenting the results concerning the increase of ESV you should refer to the specific year. In the abstract there is no information that you analyzed data for the years 2005-2018.
Response 1: We deeply appreciate the reviewer's suggestion. According to the reviewer's comment, we have added a more detailed interpretation of the specific year over which the ESV growth was described in the abstract.
Point 2: References, The article should be referenced in accordance with the editorial requirements. In the current version, for instance, you refer to Li et al. (2014) (lines 65, 68, 196…) and it is not clear which item it concerns (no 45 or 46 - lines 580-582).
Response 2: We are extremely grateful to reviewer for pointing out this problem. Consistent with the revision suggestions given by the reviewer 1, this paper does have deficiencies in the references inside the text and at the end of the paper. We modified the citations of the references as required by the journal. As for the ambiguous references involved in this paper, we have modified in accordance with the editorial requirements.
Point 3: Figures, I suggest to improve quality of some figures, especially Fig. 4.
Response 3: Thank you for underlining this deficiency. This section was revised and modified according to the information showed in the letter suggested by the reviewer. In order to enhance the readability of the article and improve the comprehensibility of the figures, we modified the legend of Figures 4, 5, 6, 7, and 8 to keep the legend of each picture consistent and facilitate comparative analysis.
Reviewer 2 Report
I found the paper very well written and presented.
Author Response
Thank you for reading my manuscript!
Reviewer 3 Report
This manuscript offers an interesting perspective to address the impact of tourism on ecosystem services value. In general, the text is well structured and organized. The manuscript provides sufficient background information and literature review regarding its topics. In general, the topics are adequately developed. The text is well structured and profusely illustrated.
Methodology
Is presented in a coherent way.
Figures
- In the spatial distribution maps the class limits must be corrected.
We can only perform comparative analysis when using the same class limits. In Figure 4 stock, for example, the authors do not take this into account. The same is true in Figure 4 increment, in Figures 5, 6, 7, and 8.
- In the graphs the information is also only comparable from having the same scale on the ordinate and abscissa axis. This is not the case. See the graphs presented in Figure 5, 6, 7 and 8.
We suggest that the authors rethink this aspect and use the same scales whenever they are using the same indicator on maps and graphs.
References
- The bibliographical references at the end of the paper must follow the journal's norms (https://www.mdpi.com/journal/sustainability/instructions#references). As an example, in the case of articles, italics are shown neither in the title of the journal nor in the number. Having said that, all references should be carefully reviewed.
- Review the references inside the text and always use the same criteria, alphabetical ordering or by year. For exemple lines 61, 62, and 63: (Qu, 2007; Buckley, 2011; Azam et 61 al, 2018; Brtnický et al., 2020; Azam et al, 2018; Wilson and Seney, 1994; Drius et al., 2019; 62 Mirzaei et al., 2020; Nagajyoti et al., 2010)
- Review all the text, sometimes are not given spaces. For exemple, Line 56: “(Zhao and Hou, 2019)and”, Line 64: “capacity(Pope et al., 2019...”
Conclusion
The authors do not point out the limitations of the research. The limitations of the study and possible future research are missing. Concentrate a bit more on current challenges in these topics. Future research perspectives should be pointed out.
Author Response
Response to Reviewer 3 Comments
Point 1:
- In the spatial distribution maps the class limits must be corrected.
We can only perform comparative analysis when using the same class limits. In Figure 4 stock, for example, the authors do not take this into account. The same is true in Figure 4 increment, in Figures 5, 6, 7, and 8.
- In the graphs the information is also only comparable from having the same scale on the ordinate and abscissa axis. This is not the case. See the graphs presented in Figure 5, 6, 7 and 8.
We suggest that the authors rethink this aspect and use the same scales whenever they are using the same indicator on maps and graphs.
Response 1:
We are grateful for the suggestion. To be more clearly and in accordance with the reviewer concerns, we have modified the legend in Figures 4, 5, 6, 7, and 8 to keep the same class limits of maps. It should be noted that in Figure 4, the author divided the ESV in 2005 into three classes and the ESV in 2018 into five classes, considering the different changes of ESV stock. In Figure 4 increments and Figures 5, 6, 7, and 8, the maps are similarly divided into several classes according to the actual situation.
We apologize for the lack of comparability of the graphs in the original manuscript. Based on R software, we unified the scale of ordinate and abscissa axis in Figures 5, 6, 7, and 8, on the basis of BRT regression model.
Point 2:
References
- The bibliographical references at the end of the paper must follow the journal's norms (https://www.mdpi.com/journal/sustainability/instructions#references). As an example, in the case of articles, italics are shown neither in the title of the journal nor in the number. Having said that, all references should be carefully reviewed.
- Review the references inside the text and always use the same criteria, alphabetical ordering or by year. For exemple lines 61, 62, and 63: (Qu, 2007; Buckley, 2011; Azam et 61 al, 2018; Brtnický et al., 2020; Azam et al, 2018; Wilson and Seney, 1994; Drius et al., 2019; 62 Mirzaei et al., 2020; Nagajyoti et al., 2010)
Review all the text, sometimes are not given spaces. For exemple, Line 56: “(Zhao and Hou, 2019)and”, Line 64: “capacity(Pope et al., 2019...”
Response 2:
We agree with the comment and standardized the reference format according to the journal's norms. Referring to the journal's website, the author carefully reviewed and corrected the bibliographical references at the end of the paper. The author unified the citation criteria of the references inside the text, and adopted the chronological order to quote the references. Furthermore, we reviewed the full text and added Spaces before parentheses for references.
Point 3:
Conclusion
The authors do not point out the limitations of the research. The limitations of the study and possible future research are missing. Concentrate a bit more on current challenges in these topics. Future research perspectives should be pointed out.
Response 3:
We are grateful for the suggestion. As suggested by the reviewer, we have added more details about defects and future research prospects in the conclusion section.
This manuscript is a resubmission of an earlier submission. The following is a list of the peer review reports and author responses from that submission.